# L-Ascorbic Acid (LAA) Supplementation as a Potential Treatment for Skin Aging: Regulation of Adipose Tissue Mesenchymal Stem Cells (AT-MSCs) Protein Secretion

**DOI:** 10.3390/cimb47060474

**Published:** 2025-06-19

**Authors:** Komang Ardi Wahyuningsih, I. Gede Eka Wiratnaya, I. Wayan Weta, I. Gde Raka Widiana, Wimpie I. Pangkahila, Ida Ayu Ika Wahyuniari, I. Made Muliarta, Veronika Maria Sidharta, Assyafiya Salwa

**Affiliations:** 1Doctoral Program, Faculty of Medicine, Udayana University, Denpasar, Bali 80232, Indonesia; komang.wahyuningsih@atmajaya.ac.id (K.A.W.); eka.wiratnaya@unud.ac.id (I.G.E.W.); wy_weta@unud.ac.id (I.W.W.); raka_widiana@unud.ac.id (I.G.R.W.); wimpie.pangkahila@unud.ac.id (W.I.P.); ikawahyuniari@unud.ac.id (I.A.I.W.); made.muliarta@unud.ac.id (I.M.M.); 2Department of Histology, School of Medicine and Health Sciences, Atma Jaya Catholic University of Indonesia, North Jakarta, Jakarta 14440, Indonesia; 3Faculty of Medicine, UPN Veteran Jakarta, Jakarta 12450, Indonesia; assyafiyasw@gmail.com; 4Metabolic Disorder, Cardiovascular and Aging Research Center, Indonesian Medical Education and Research Institute, Faculty of Medicine Universitas Indonesia, Central Jakarta, Jakarta 10430, Indonesia

**Keywords:** adipose tissue mesenchymal stem cells (AT-MSCs), L-ascorbic acid (LAA), reactive oxygen species (ROS), skin aging

## Abstract

Skin aging is mostly caused by the accumulation of reactive oxygen species (ROS) that lead to cellular dysfunction. One promising therapy for skin aging is the secretome product of adipose tissue mesenchymal stem cells (AT-MSCs). L-ascorbic acid (LAA) is an essential molecule for preventing oxidative stress as an external antioxidant agent and has been used in chemical therapy for skin aging. In this study, we evaluated the effects of LAA on cell morphology, the number of cells, cell viability, and the paracrine secretion of preconditioned AT-MSCs in in vitro culture with LAA in 100 and 200 µg/mL compared with an untreated culture with LAA as a control. LAA supplementation in both concentrations improved the morphology of cells without affecting the cell viability. However, there was no significant improvement in the number of cells even though the trend showed an enhancement of the number of cells. The total protein of the secretome decreased in the LAA preconditioning group. However, preconditioning AT-MSCs in in vitro culture with LAA improved the levels of insulin-like growth factor 1 (IGF-1), transforming growth factor β1 (TGF-β1), and interleukin 6 (IL-6) which are essential proteins for skin aging in regulating ROS.

## 1. Introduction

Skin is the outer organ of the human body that has direct contact with multiple environmental factors, such as ultraviolet lights and pollution, and is affected by internal factors, like disease and the deterioration of structural and physical function. The molecular mechanisms of skin aging are mostly caused by the accumulation of reactive oxygen species (ROS) that lead to cellular dysfunction [1,2].

Cells produce endogenous antioxidants, such as superoxide dismutase (SOD), glutathione peroxidase (GSH-Px), and catalase (CAT), as a response to oxidative stress. However, the defense system of cells declines along with aging [2]. L-ascorbic acid (LAA) is a potent external antioxidant that reduces oxidative stress and protects cellular health. This molecule protects cells from ROS by electron transfer to inhibit oxidation and support endogenous antioxidant enzymes. LAA also protects macromolecules from the damage caused by external exposure, such as pollutants [2,3]. LAA is known to increase collagen production through the upregulation of collagen gene expression [3].

In terms of cellular therapy, mesenchymal stem cell (MSC) therapies have a great impact on skin regeneration and rejuvenation. MSCs are multipotent stem cells that can be collected from many sources, including bone marrow, adipose tissue, umbilical cord, peripheral blood, muscles, skin, etc. MSCs produce various types of bioactive components, namely growth factors, cytokines, chemokines, lipids, hormones, extracellular vesicles, and miRNA, with broad therapeutic properties, such as immunomodulator, angiogenesis, anti-apoptotic, anti-fibrotic, chemo-attraction, antioxidant, etc. [4]. The collective component that is secreted by MSCs is known as secretome.

Adipose tissue is one of the common MSC sources for therapy because it is reliable and easy to obtain from the byproducts of the esthetic medical procedure, liposuction [5]. Studies have shown that AT-MSCs can improve skin wrinkles caused by photoaging by stimulating collagen synthesis in human dermal fibroblasts (HDFs), inhibiting telomere shortening and estrogen depletion, reducing intracellular ROS levels and preventing their excessive production, enhancing mitochondrial numbers, and reducing apoptosis caused by ultraviolet B (UVB) exposure [4].

MSCs have the ability to adapt to the microenvironment and change their phenotype and functions so that the secretome will vary following the culture conditions [5,6]. Based on its plasticity, MSC preconditioning with specific molecules influences the secretion of MSC secretome. The effect of preconditioning AT-MSCs in in vitro culture with LAA to enhance the production of specific antioxidant bioactive molecules remains unclear. In this study, we evaluate the effects of LAA on cell morphology, the number of cells, cell viability, and the paracrine secretion of preconditioned AT-MSCs in in vitro culture with LAA in 100 and 200 µg/mL compared with untreated cells with LAA as a control. The composition of the secretome of AT-MSC preconditioning with LAA will be characterized based on total protein and the levels of insulin-like growth factor 1 (IGF-1), transforming growth factor β1 (TGF-β1), and interleukin 6 (IL-6). IGF-1 is a growth factor that can inhibit DNA damage in aging dermal fibroblasts [7]. TGF-β1 is reported to stimulate the synthesis of the extracellular matrix and type 1 procollagen and suppress tyrosinase, which plays a role in melanin synthesis, thereby preventing dermal thinning and hyperpigmentation [7,8]. The level of IL-6 in the secretome needs to be measured as it contributes to reducing oxidative stress by enhancing the activator of transcription 3 (STAT3), nuclear factor erythroid 2-related factor 2 (Nrf2), and SOD [8].

## 2. Materials and Methods

### 2.1. Isolation and Culture of AT-MSCs

AT-MSCs were isolated from the subcutaneous adipose tissue of patients undergoing abdominal liposuction under approval from the ethics committee from the School of Medicine and Health Sciences, Atma Jaya Catholic University of Indonesia, with ethics number 17/06/KEP-FKIKUAJ/2023. Following extraction, a transport medium consisting of low-glucose (100 mg/dL) Dulbecco’s Modified Eagle’s Medium (DMEM) (Gibco, Grand Island, New York, NY, USA) with 4 mM L-glutamine (Gibco, Grand Island, New York, NY, USA) and 1% antibiotic–antimycotic solution (penicillin 10,000 units/mL streptomycin 10,000 mg/mL (Gibco, Grand Island, New York, NY, USA)), and amphotericin B 25 mg/mL(Gibco, Grand Island, New York, NY, USA)) was added to the tube containing the lipoaspirate (adipose tissue mixed with tumescent fluid). The tube was stored in a cool box with ice packs (maintaining a temperature from 8 °C to 4 °C) and transported to the laboratory for processing. The isolation of AT-MSCs was performed according to the protocol by Pawitan et al. [9]. Initially, the adipose tissue was separated from the tumescent fluid using a coffee filter and washed with sterile 1X phosphate-buffered saline (PBS) at pH 7.4 until the adipose tissue was free of blood. The tissue was then placed into a sterile 50 mL centrifuge tube, mixed with a 0.075% collagenase type I solution (Sigma, Darmstadt, Germany), incubated in a CO_2_ incubator for 1 h at 37 °C, and agitated every 5 min.

After incubation, the liquid phase (infranatant) was removed with a serological pipette, transferred to a sterile 15 mL centrifuge tube, and centrifuged for 10 min at 1200 rpm. The liquid was then aspirated with a serological pipette, leaving only the cell pellet (stromal vascular fraction, SVF) at the bottom of the tube. The SVF was resuspended in a mixture of low-glucose DMEM with L-glutamine, 10% human serum, and 1% sterile antibiotic–antimycotic solution (filtered with a 0.2 µm filter), referred to as the complete culture medium. The cells were then seeded in a 12-well plate (5000 cells per cm^2^ well) and incubated in a 5% CO_2_ incubator at 37 °C.

After 2 or 3 days, the cells were observed, and the culture medium was changed every 2−3 days. The cells were monitored until fibroblast-like plastic adherent cells were seen attaching to the bottom of the 12-well plate. Once the cells reached 70−80% confluence, they were harvested by replacing the culture medium with 1X PBS at pH 7.4 (Gibco, Grand Island, New York, NY, USA), rinsing the cells twice, and adding TripLE^TM^ Select Enzyme (Gibco, Grand Island, New York, NY, USA) to the 12-well plate to detach the cells from the plate. The cells were observed under an inverted microscope (Nikon Eclipse Ti-S). Once all cells had detached from the plate, the complete culture medium was added to the 12-well plate to stop the enzyme dissociation reaction. The cells were resuspended, transferred to a sterile 15 mL centrifuge tube, and centrifuged for 10 min at 1200 rpm. The cell pellet was then resuspended in the complete culture medium for expansion or subculture (passaging) and then stored in an N_2_ tank (−196 °C) for further use. AT-MSCs from passage 4 and passage 7 were used for LAA supplementation treatment in 100 and 200 µg/mL and control treatment (without LAA).

### 2.2. Characterization of AT-MSCs

The cell culture at passage 3 was characterized according to the standard minimum criteria of MSCs by the International Society for Cell & Gene Therapy (ISCT) [10], which are cell morphology, immunophenotype expression, and the potential of cell differentiation.

#### 2.2.1. Cell Morphology Analysis of AT-MSCs

Once AT-MSCs confluency reached 80% P3, cell morphology was assessed under an inverted microscope and documented. Then some cultures were treated with a differentiation medium for the analysis of cell differentiation and the others were enzymatically detached as described and counted. The cell pellets were resuspended with a complete medium and then counted with Trypan Blue Solution, 0.4% (Gibco, Grand Island, New York, NY, USA) staining in a haemocytometer under an inverted microscope. The cell suspension was then used for immunophenotyping and sub-culturing for LAA treatment.

#### 2.2.2. Immunophenotypic Analysis of AT-MSCs

MSC surface markers were measured with BD Stemflow™ Human MSC Analysis Kit (BD Bioscience, Franklin Lakes, NJ, USA) in a flow cytometer which assesses the percentage of cells with CD90, CD73, CD105, and Lin surface markers. In brief, cells at passage 3 were resuspended in sterile 1X PBS at pH 7.4 and then transferred to the Falcon tube at 1 × 10^5^ cell/100 µL. The suspension was divided into three different tubes, which are stained, unstained, and isotype. Positive (CD90 FITC, CD105 PerCP-Cy™5.5, and CD73 APC) cocktail and negative (CD34 PE, CD11b PE, CD19 PE, CD45 PE, and HLA-DR PE) cocktail reagents were added to the stained tube, positive and negative isotype reagents were added to the isotype tube, and no reagent was added to the unstained tube. All tubes were incubated in the dark, at room temperature for 30 min. The samples were resuspended with 100 µL sterile 1X PBS at pH 7.4 and then assessed with a fluorescence-activated cell sorting (FACS) machine (BD FACSAria III; BD Bioscience). The data was obtained as a percentage of each marker.

#### 2.2.3. Differentiation of AT-MSCs

Passage 3 of AT-MSCs from donors at 70–80% confluence were harvested and washed with PBS 1X pH 7,4 twice. Then, we resuspended the cells in a completely cultured medium (refer to the Isolation and Culture of AT-MSCs Section) and placed them in a 24-well plate (2 × 10^5^ cells per well). After 70–80% confluence, the medium was changed to an induction medium that was adipogenic and osteogenic, as instructed by the kit manufacturer, which is StemPro™ Osteogenesis Differentiation Kit and StemPro™ Adipogenesis Differentiation Kit (Thermo Fisher Scientific, Waltham, MA, USA). For the chondrogenic differentiation, cells were kept in prolonged culture in a complete culture medium. The observation of cell culture was performed using an inverted microscope. Cell differentiation happened during the alteration of cell morphology after 4 days of induction. Oil red O, Alcian blue, and Alizarin red staining chemicals were used to identify the differentiation of cells into adipocytes, chondrocytes, and osteocytes, respectively.

### 2.3. Preconditioning of AT-MSC Culture for Secretome Collection

The production of AT-MSC secretome was carried out following the protocol from Prakoeswa et al. [11], with modifications to the type of antioxidant supplementation. The secretome preconditioning was completed in passages 4 and 7 of the culture. Specifically, AT-MSCs at passage 3 from each donor, which had reached 70−80% confluence, were cultured in the starvation condition of complete culture medium, with a reduced concentration of human serum (2%), then was mixed with LAA at 100 and 200 µg/mL. The concentration of LAA was based on our previous research (unpublished). The control group was cultured without supplementation of LAA. The medium was added to wells that had been rinsed with 1X PBS at pH 7.4. The cells were then incubated at 37 °C with 5% CO_2_ for 48 h. After 48 h, the medium was collected and filtered using a 0.45 µm syringe membrane. Before analysis, samples were stored at −80 °C until all samples were collected.

### 2.4. Testing the Effects of L-Ascorbic Acid on AT-MSC Culture and Secretion

The next step after preconditioning was to evaluate the effects of LAA on cell morphology, cell number and viability, and secretome profile.

The cells were subcultured until passages 4 and 7 in a flask culture for the LAA and control treatment. After the cells reached confluency at 70−80%, the cells were rinsed with sterile 1X PBS at pH 7.4 two times. AT-MSC culture morphology was assessed using an inverted microscope and documented. The cells then detached as described above. Cell pellets were resuspended with a complete medium and then counted with Trypan Blue Solution, 0.4% (Gibco^TM^) staining in a haemocytometer under an inverted microscope. Cell viability was calculated with the formula:$$Cell\;viability=\frac{Number\;of\;live\;cells}{Total\;cells\;}\times 100\%$$

The evaluation of AT-MSC secretion was based on total protein by BCA Protein Assay (FineTest, Wuhan, China) the levels of IGF-1, TGF-β, and IL-6 by Enzyme Linked Immunosorbent Assay (ELISA) with ELISA kit (FineTest, Wuhan, China).

### 2.5. Statistical Analysis

GraphPad Prism 10.1.0 (GraphPad Software Inc., Boston, MA, USA) was applied for data analysis. Shapiro–Wilk’s test was performed to check the normality of data distribution. One-way ANOVA was employed for comparison between normal distributed data, followed by Tukey’s multiple test and the data was expressed in mean ± SD. Kruskal–Wallis’ test was employed for comparisons between not normal distributed data, followed by Dunn’s multiple test and the data was expressed as median values with interquartile range. *p* < 0.05 was indicative of a statistically significant difference.

## 3. Results

### 3.1. AT-MSC Characterization

AT-MSC culture morphology, in general, displayed typical adherence and were fibroblastic shaped (Figure 1). According to the MSC surface markers measured by flow cytometry, the AT-MSC culture at passage 3 expressed positive markers of MSCs, which are 99.8% of CD73, 95% of CD90, and 62.7% of CD105, and negative markers of MSCs, which are 0.1% of the types of hematopoietic cell surface markers (CD34, CD11b, CD19, CD45, and HLA-DR) (Figure 2).

AT-MSC differentiation is depicted in Figure 3. Figure 3a shows that the AT-MSCs have differentiated into chondrocytes, as seen by the change in AT-MSCs morphology from fibroblast-shaped to rounded, polygonal, blue-stained chondrocytes [12]. Figure 3b below shows the AT-MSCs’ differentiation into osteocytes, as seen by the appearance of the red-stained calcium deposits on the surface of the culture [13]. Figure 3c shows the AT-MSCs’ differentiation into adipocytes, detected as an accumulation of red-stained intracellular lipid droplets [14].

### 3.2. LAA Treatment in AT-MSCs: Improved Cell Morphology Without Affecting Cell Viability

The inclusion of LAA in the cell culture medium significantly improved the cell morphology of the AT-MSC culture compared with the control, as shown in Figure 4. The morphology of the cells supplemented with LAA was flat and elongated in shape with a smaller size than the control. The control cells looked like they had a bigger cell size with an oval shape. LAA supplementation at 200 µg/mL in AT-MSC passage 7 significantly showed an enhancement of total live cells compared with the control group in passage 4. There was no significant difference in the total live cells in other groups compared with the control group. However, the trend showed that the total live cells of the LAA supplementation groups were increased compared with the control group from each passage, as the LAA concentration and the number of passages increased (Figure 5). LAA did not significantly affect cell viability (compared with the control group from each passage), with the % of viability for each group being around 83–100% (Figure 5).

### 3.3. Effect of LAA Treatment on AT-MSC Protein Profile and Secretion

This conducted research showed that the total protein concentration of AT-MSC secretome in passages 4 and 7 was significantly decreased in the LAA 100 and LAA 200 groups compared with the control group from each passage. However, there was no significant difference between the LAA 100 and LAA 200 groups in passage 7 (Figure 6).

In contrast, the LAA 100 and 200 µg/mL groups of AT-MSC passage 4 in the secretion of IGF-1, TGF-β1, and IL-6 showed fluctuated values compared with the control group from each passage. The best concentration of IGF-1 is shown in the LAA 200 group of passage 4. The IGF-1 concentration of the LAA 200 group in passage 4 was significantly higher than the LAA 100 group in passage 4 and the LAA 200 group in passage 7. The LAA 200 group in passage 4 also showed a higher concentration than the control group but did not reach statistical significance due to the variation among the replications. There was no significant difference in the LAA 100 group and the LAA 200 group compared with the control group in passage 7 (Figure 7).

The concentration of TGF-β1 in the LAA 100 group and the LAA 200 group of passage 4 was slightly higher than the control group. However, these did not show any significant differences. The LAA 200 group of passage 7 was significantly higher than the control group and not significantly higher than the LAA 100 group. There was no significant difference between the LAA 100 group and the control group in passage 7. The concentration of IL-6 in the LAA 200 group of passages 4 and 7 was significantly higher than the control group from each passage. However, it did not reach the statistical significances of the LAA 100 group in passages 4 and 7 compared with the control group from each passage. IL-6 concentrations showed significant differences between the LAA 100 group and the LAA 200 group of passages 4 and 7; the LAA 200 group had a higher concentration than the LAA 100 group. From the comparison between the AT-MSC passages 4 and 7 groups, the concentration of IGF-1 and TGF-β1 in each group did not differ significantly, while the concentration of IL-6 tended to decrease.

## 4. Discussion

Mesenchymal stem cell (MSC) secretome is a promising alternative that has been widely used for regenerative medicine, including rejuvenation. MSC secretome contains various components that are necessary for anti-aging and skin regeneration [15].

Cellular oxidative stress caused by the accumulation of reactive oxygen species (ROS) is considered as one of prominent factors of skin aging, as the result of progressive mitochondrial dysfunction and external factors like UV radiation. Excessive ROS production leads to further cellular damage and senescence because of the deterioration of cellular biomolecules, including DNA, proteins, and lipids. The most common sign of skin aging that is affected by the enhancement of ROS is collagen reduction, which causes the formation of wrinkles, dermis thinning, skin fragility, the loss of elasticity, and pigmentary change [16].

To improve MSCs’ therapeutic effect related to ROS regulation in skin aging, a modification was made by priming MSCs with small molecules like L-ascorbic acid (LAA), also known as vitamin C. LAA is an antioxidant with an inhibitory role in ROS production [17] and it alters the cell culture microenvironment so it has been used in supplementation for cell cultures and cell differentiation [18].

Fujisawa et al. [19] noted some improvements in MSC culture with LAA supplementation, namely cell growth and proliferation, cell survival, cell cycle progression, and DNA replication and repair.

The AT-MSC characteristics of this study have met the minimum criteria of MSCs by the ISCT for cell morphology and differentiation potential [10]. The cells were adhered to plastic cultures with a fibroblast shape and able to differentiate into osteoblasts, adipocytes, and chondroblasts according to in vitro staining. The percentage of positive and negative MSC surface markers of the culture were in accordance with the minimum criteria for defining MSCs by the ISCT, which are ≥95% positive (CD73 and CD90) and ≤2% negative (CD45, CD34, CD14, CD19, and HLA-DR) [10]. Our findings show that the AT-MSCs expressed positive markers of CD73 and CD90 and were negative for hematopoietic cell surface markers (CD34, CD11b, CD19, CD45, and HLA-DR). However, the expression of the CD105 surface marker had a lower expression than the minimum percentage from the ISCT. MSC heterogeneity from different cultures has been studied, especially for the expression of CD105 (Endoglin). Its expression level has a significant effect on MSC characteristics. MSC populations in high and low CD105 have no difference in cell growth. An MSC population in low CD105 was more likely to undergo osteogenic and adipogenic differentiation [20] but did not correlate with chondrogenic differentiation [21]. Low-CD105 MSC populations have a higher effect on CD4^+^ T cell suppression than high CD105 [20].

This study has revealed that LAA in a cell culture medium significantly improved the cell morphology of the AT-MSC culture. The morphology of the AT-MSC culture indicated that the cell size of the control group was larger compared with the LAA groups. Wahyuningsih et al. [22] demonstrated that LAA in AT-MSCs from diabetic donors could maintain the proper size of cells compared with the control group without LAA that have the highest spread size. A higher spread size of cells is one of the signs of cellular senescence. This study has shown that LAA in a cell culture medium causes the improvement of AT-MSC proliferation without affecting their viability. A study by Segeritz et al. explained that a % viability around 80–95% is categorized as good viability [23]. The possible mechanism is that LAA upregulates cyclin E1 and CDK2 with the downregulation of p53 and p21 as the mechanism to enhance cell proliferation and the number of cells in the S and G_2_/M phase [24]. The ability of LAA to enhance cell proliferation is dose-dependent. A higher dose of LAA has a potency to promote oxidation and ATP starvation [19] and leads to the cytotoxicity of cells by decreasing cell viability [25]. Our research has shown there was no significant difference between the LAA 100 and 200 µg/mL groups in terms of improving the cell proliferation. However, both concentrations were safe for cell viability.

MSC secretome contains various components that are necessary for anti-aging and skin regeneration, such as IGF-1, TGF-β1, and IL-6. This study showed that AT-MSCs enhanced the secretion of IGF-1, TGF-β1, and IL-6 in the presence of LAA. These findings are consistent with research by Bhandi et al. [18] that LAA treatment at 10 µM, 25 µM, 50 µM, and 100 µM doses increases the secretion of IGF-1 and IL-6 by MSCs from Human Exfoliated Deciduous Tooth (SHEDs) compared with a control. LAA also increases TGF-β, in the specific isoform latent TGF- β1, during the release and activation in mammalian cells [26].

LAA treatment in cells has been proven to upregulate IGF-1 expression and secretion through the PI3K pathway as it promotes keratinocytes proliferation [27]. LAA, as a cofactor for hydroxylase, an enzyme for collagen fiber formation, induces the upregulation of collagen type 1 A (Col1A), collagen type 5 A1 (Col5A1), and α2β1 integrin through the ERK1/2 pathway [18,19]. The administration of LAA to cells can enhance (pro)collagen deposition, which is further increased with the addition of TGF-β1 [28].

MSC secretome also normally contains IL-6 and tends to be high compared with other interleukins [29]. IL-6 is referred to as a pleiotropic cytokine, acts as a pro-inflammatory cytokine by regulating immune response (B-cell maturation and T-cell survival), and acts as an anti-inflammatory cytokine by inducing the secretion of anti-inflammatory IL-10 and the IL-1 receptor antagonist, promoting a macrophage switch from M1 pro-inflammatory to M2 anti-inflammatory and reducing TNF-α [30,31]. The increase in IL-6 level can be caused by a low glucose starvation condition [32] to protect cells from apoptosis due to the starvation conditions during the pre-conditioning of AT-MSC secretome collection as the inflammation response. LAA is supposed to decrease IL-6 expression due to the inhibition of the pro-inflammatory signaling operated by NF-κB according to Gegotek et al. [33]. This is contrary to Bhandi et al. [18] and our research result that LAA has the effect of enhancing IL-6 secretion in MSCs. IL-6 expression could be activated by the phosphorylation of ERK1/2 in the mitogen-activated protein kinases (MAPK) signaling pathway [34]. Based on Langenbach and Handschel [35], LAA regulates the phosphorylation of ERK1/2 in the MAPK signaling pathway. The expression of IL-6 is also induced by the presence of TGF-β1 through the MEK signaling pathway [36].

IGF-1, TGF-β1, and IL-6 are essential molecules that are necessary for anti-aging related to oxidative stress. IGF-1 relieves premature senescence caused by H_2_O_2_ through the inhibition of the p53-progerin pathway [37] and regulates the expression and activity of internal antioxidant enzymes, such as catalase, SOD, and glutathione peroxidase (GPx) [38]. TGF-β1 stimulates the production of collagen, which is part of the extracellular matrix (ECM), through the TGF-β/Smad signaling cascade [16,28]. By repairing the ECM environment in aged skin, it is possible to reduce further oxidative stress as a study showed the ECM’s ability to protect cells from free radicals by significantly reducing oxidative marker expression [39]. IL-6 is essential for skin anti-aging by enhancing the proliferation of keratinocytes through the activation of the STAT-JAK signaling pathway. IL-6 also binds to IL-6Rα to induce collagen deposition by upregulating TGF-β1 in fibroblasts, and inducing angiogenesis by enhancing VEGF secretion, macrophage, and neutrophil infiltration [30]. In oxidative stress regulation, IL-6 promotes internal antioxidant Nrf2 to use mitochondria to reduce ROS production and protect cells from pro-inflammatory cytokines related to apoptosis [40]. However, further investigation needs to be performed to evaluate the safety of IL-6 in MSC secretome for skin therapy and its potential to promote the inflammation process as a pro-inflammatory cytokine.

Further studies are required to identify other proteins related to skin aging therapy and potential proteins that are involved in skin aging in secretome. The mechanism of regeneration and the biological function of secretome also need to be examined to confirm its potential for skin regeneration and rejuvenation. The replication/sample size of the research also needs to be improved.

## 5. Conclusions

The supplementation of LAA in AT-MSC cultures has been demonstrated to delay cellular aging. The morphology improved in both LAA concentrations compared with the control without affecting cell viability. LAA 200 µg/mL had the effect of enhancing the number of cells in higher passages with the trend of total live cells with LAA supplementation increasing as the LAA concentration and the number of passages increased. The total protein of the secretome slightly decreased in the LAA preconditioning group. However, preconditioning AT-MSCs in in vitro culture with LAA improved the levels of IGF-1, TGF-β1, and IL-6, which are essential proteins for skin aging. LAA-treated AT-MSC secretome may provide newer treatment strategies for skin regeneration and rejuvenation. Further examination needs to be conducted to identify other proteins related to skin aging therapy and the potential proteins that are involved in skin aging in the secretome, proving the potential of LAA-treated AT-MSC secretome in skin aging and understanding its mechanisms of regeneration.

## Figures and Tables

**Figure 1 cimb-47-00474-f001:**
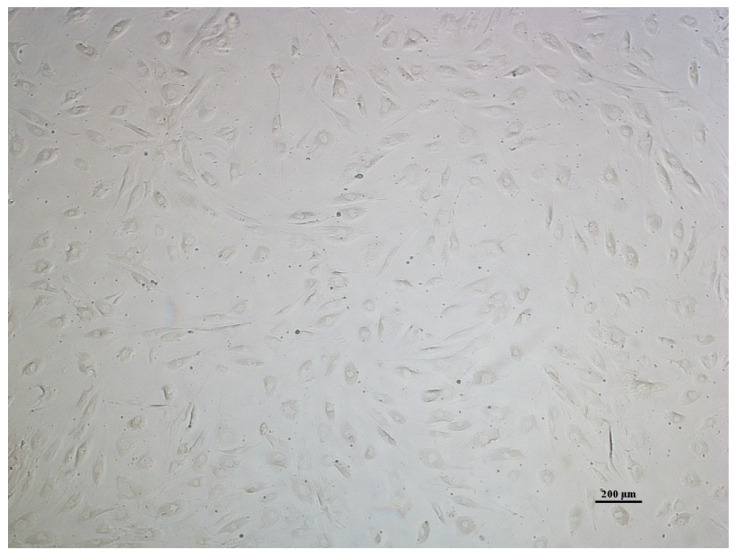
Morphology of AT-MSCs in passage 3, magnification 40×.

**Figure 2 cimb-47-00474-f002:**
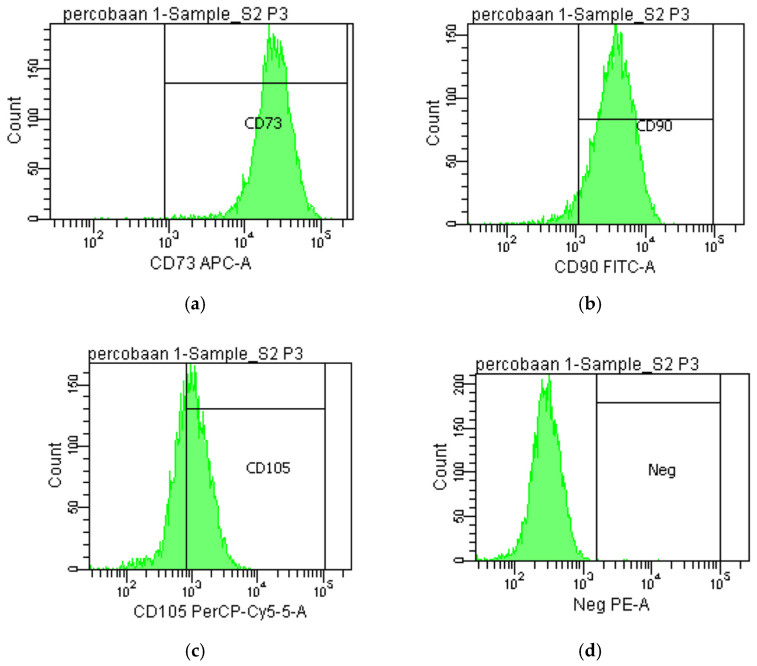
The expression of AT-MSC surface markers of (**a**) CD73, (**b**) CD90, (**c**) CD105, and (**d**) negative markers in passage 3 (P3). This experiment was performed one time with no replication (n = 1).

**Figure 3 cimb-47-00474-f003:**
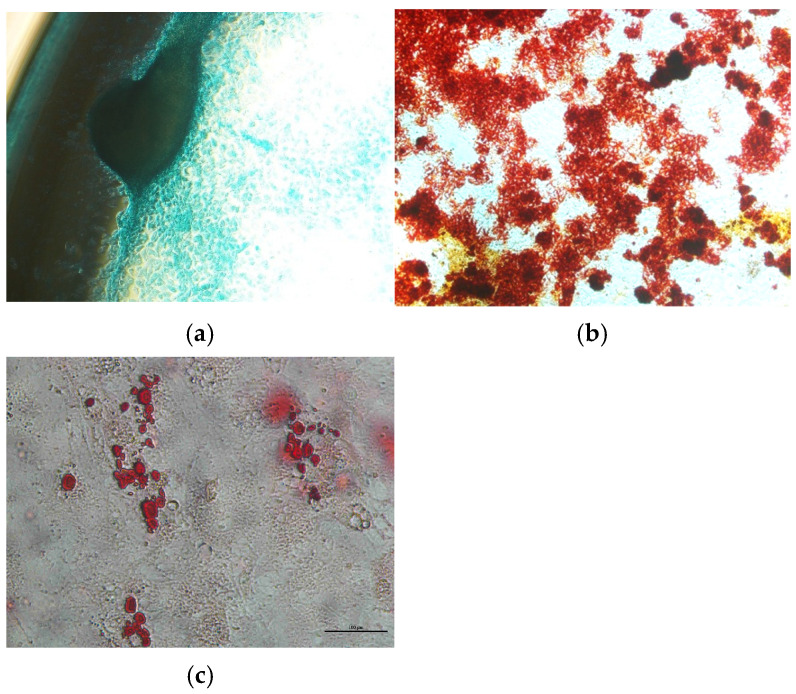
The AT-MSCs’ differentiation into (**a**) chondrocytes, (**b**) osteocytes, and (**c**) adipocytes at passage 3; magnification 40×.

**Figure 4 cimb-47-00474-f004:**
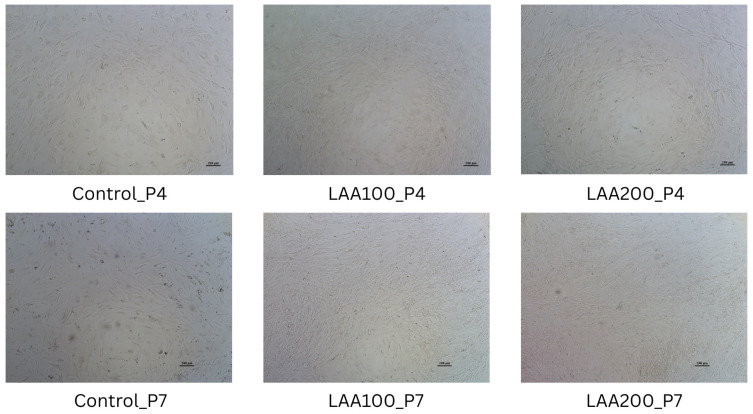
Morphology of AT-MSCs in passage 4 (P4) and 7 (P7), magnification 40×.

**Figure 5 cimb-47-00474-f005:**
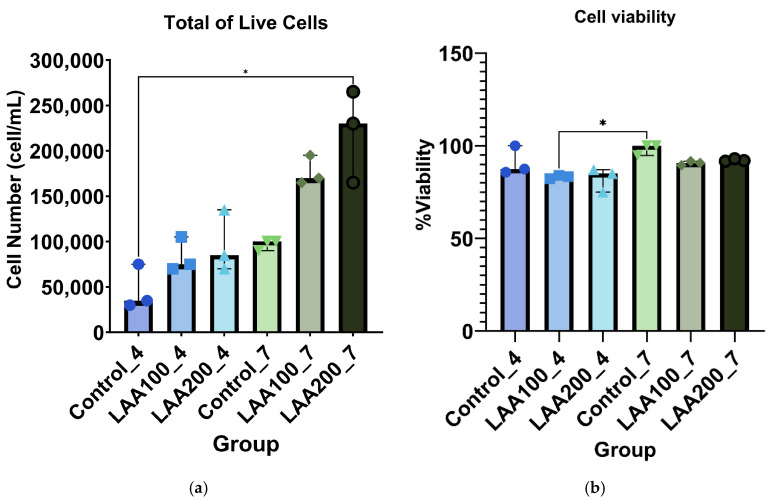
AT-MSC (**a**) number of live cells and (**b**) viability in passages 4 and 7 for control, LAA 100 µg/mL, and LAA 200 µg/mL groups; median with 95% CI of n = 3; * *p* < 0.05 indicates statistical significance.

**Figure 6 cimb-47-00474-f006:**
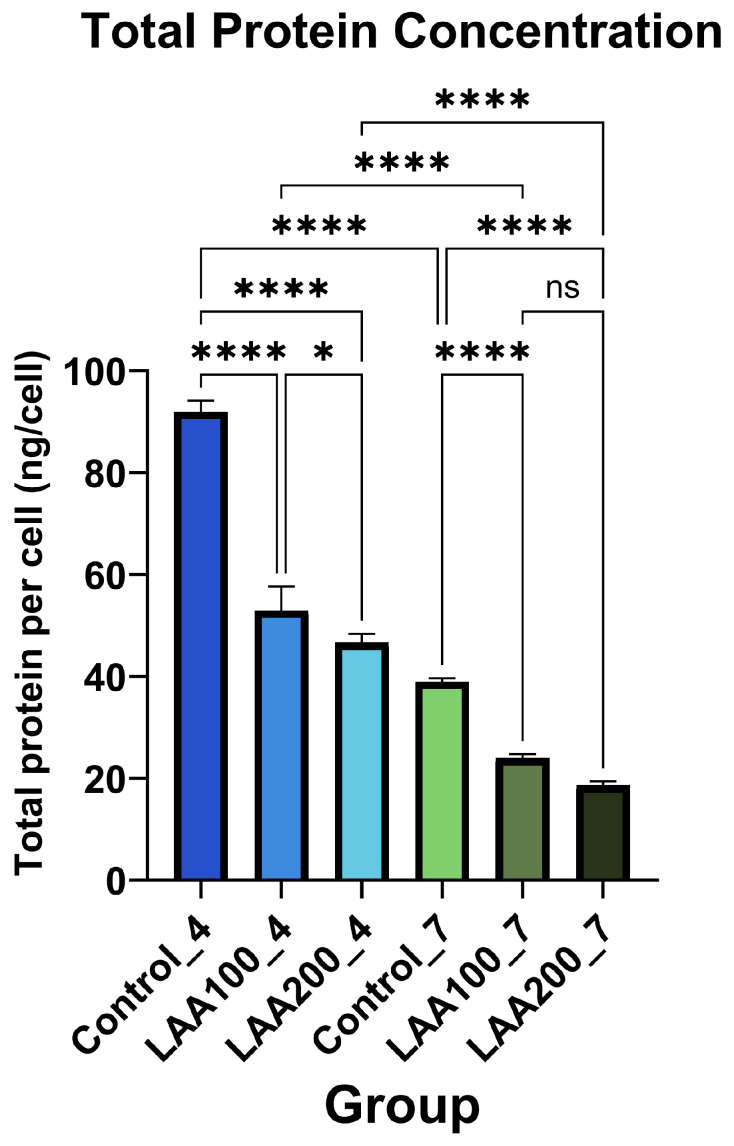
Total protein concentration of AT-MSC secretome; means ± SD of n = 3; *, **** *p* < 0.05 indicates statistical significance and ns *p* > 0.05 indicates statistical insignificance.

**Figure 7 cimb-47-00474-f007:**
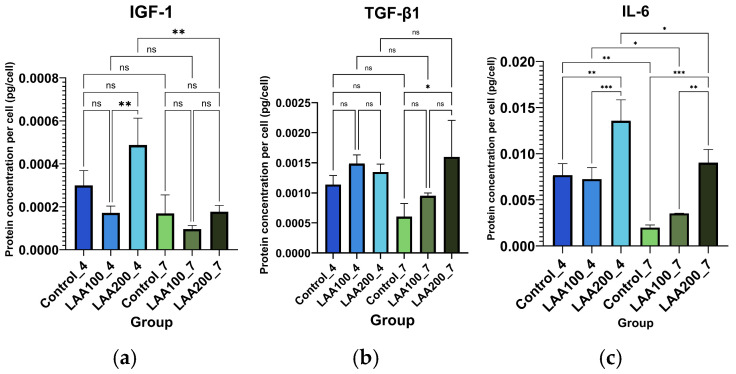
Growth factor and cytokine profiles of AT-MSC secretome: (**a**) concentration of IGF-1; (**b**) concentration of TGF-β1; and (**c**) concentration of IL-6; means ± SD of n = 3; *, **, *** *p* < 0.05 indicates statistical significance and ns *p* > 0.05 indicates statistical insignificance.

## Data Availability

No datasets were generated or analyzed during the current study.

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
