# Peer review of "L-Ascorbic Acid (LAA) Supplementation as a Potential Treatment for Skin Aging: Regulation of Adipose Tissue Mesenchymal Stem Cells (AT-MSCs) Protein Secretion"

_cimb, 2025, doi:10.3390/cimb47060474_

Round 1
Reviewer 1 Report
Comments and Suggestions for Authors
General comments
The manuscript describes the effect of L-Ascorbic Acid (LAA) treatment on Adipose-Tissue Mesenchymal Stem Cells (AT-MSCs). It attempts to show that LAA increases the proliferation of AT-MSCs and the secretion of proteins that are known to reduce skin aging. However, the effects of LAA on cell number was not significant and thus the sentence in the abstract, line 20, “number of cells have improved in both LAA concentrations” is not justified. This means that the only important (and significant) results of this manuscript is the effect of LAA on the secretome of AT-MSCs. These findings are novel for the specific type of MSC but similar results were shown for other types of MSC as cited in reference 18. The title is also not appropriate as it includes the phrase "as a Potential Treatment for Skin Aging" which is not justified by the results presented in the manuscript as experiments on skin or skin cells were not performed. The English of the title is not clear and confusing. A more appropriate title may be “L-Ascorbic Acid treatment of Adipose-Tissue Mesenchymal Stem Cells (AT-MSCs) increase the secretion of proteins that can reduce skin aging”.
The description of results suffers from many problems as detailed in the specific comments.
specific comments.
- Title: see comment above.
- lines 20-21: “number of cells have improved in both LAA concentrations”. The effects of LAA on cell number in both passages are not significant and thus this sentence is not justified.
- lines 81-82: “The tube was stored in a cool box with ice packs (maintaining a temperature of 8°C to −20°C)”. There is a big difference between these two temperatures. Live cells usually should be kept at temperature above zero. −20°C can lead to freezing and death of the cells. Please specify the temperature of the tube, not of the ice packs.
- line 124: Should be “as a percentage of cells with each marker”
- lines 167-168: “viability was counted with the formula” change to: viability was calculated with the formula:
- line 187: The results of % cells with each marker should be presented ±SD and the number of experiments performed should be specified. If only one experiment was performed, this should be specified.
- lines 209-211: These two claims are not supported by the statistical analysis. There is no statistical difference between the control and the LAA treated cells in both passages and no statistical difference between control 4 and control 7.
- lines 218-219, 247-249: There are no legends to figures 5, 6 and 7. The legend should specify the number of cells seeded, the number of experiments done and the number of samples in each experiment. Without it, the SDs shown in the figures and the statistical analysis cannot be evaluated.
- lines 222-224: The results are presented as “mg/ml” but according to Fig 5 the number of cells is higher in passage 7 than in passage 4 and thus it is incorrect to compare the mg protein which is secreted by different number of cells. The right comparison is “mg/cell” – mg protein divided by the number of cells. The same calculations should be made in Figs. 6 and 7.
- lines 226-227: The same phrase appears twice.
- lines 228-236: All these results are presented in Fig 7 and there is no need to repeat the values in the text. The authors should describe the results according to Fig 7 and relate ONLY to the statistically different values within the same passage.
- 226-228: In passage 4 only IL-6 at LAA200 shows a statistical difference. Thus, to write that all 3 proteins at both concentrations “were relatively increased compared to control” is incorrect.
- lines 236-237: The non-significant decrease in IGF-1 for LAA100 compared to the control is true only for passage 4 and not for passage 7 where the values are very similar.
- lines 237-239: What is the importance of the increase in IGF1 between LAA100 and LAA200 if none of them is different from the control?
- lines 239-241: There are no statistical differences in TGFβ between LAA100 and LAA200, so either write that the increase is not significant or ignore this result.
- Figure 7 b and c: The increase in TGFβ (P7) and IL6 (P4 and 7) by LAA is not described in the text. It is not clear why the differences between LAA 100 and 200 are described but not the difference from the control.
- lines 253-259: The mechanisms discussed in this paragraph have nothing to do with the current manuscript that did not check the effects of the secretome on skin cells. These lines should be removed.
- lines 281-284: These lines repeat the results that were presented in lines 187-189. The results should only be discussed here without this repetition
- lines 292-297: The effect of LAA on AT-MSCs morphology was already published by the same group (ref 22) and it cannot be described as a novel finding in the current study.
- lines 318-319: This sentence is incorrect. Reference 28 does not show that collagen upregulates TGFβ expression.
- line 358: The English it is not clear, but if this sentence is a conclusion form the current manuscript, this conclusion is incorrect as no experiments of skin aging were done.
- lines 359-360: The conclusion “number of cells have improved in both LAA concentrations compared to control” is incorrect because there are no statistically significant differences between LAA and control.
Comments on the Quality of English Language
The Language of the manuscript should be edited by an English-speaking professional as many sentences are not completely clear
Reviewer 2 Report
Comments and Suggestions for Authors
Make some comment limitations that this has limits for extrapolations about use in beauty creams
Author Response
Comment: Make some comment limitations that this has limits for extrapolations about use in beauty creams
Response: Thank you for pointing this out. The application of LAA-preconditioned AT-MSCs secretome for skin aging treatment can be done in several ways, such as dermal injection and topical. In topical administration, secretome is usually combined with scaffold/carrier, for example, hydrogel (Clinical Trial: 0052/LOE/302.4.2/VII/2020). Therefore, using secretome in beauty products such as cream or serum is possible.
Reviewer 3 Report
Comments and Suggestions for Authors
The authors present an interesting and relevant study addressing the role of L-ascorbic acid (LAA) in modulating the secretome of AT-MSCs in the context of skin aging, an area with promising therapeutic implications.
However, to improve the scientific clarity and overall quality of the manuscript, several important revisions are required:
- Abstract
- The abstract needs to be rewritten. Most notably, it lacks a clear conclusion and is currently more descriptive than informative. Please include a brief and specific concluding statement summarizing the main findings and their implications.
- English Language and Clarity
- The manuscript requires a thorough English language revision. Several sentences are unclear or grammatically incorrect, which affects readability and comprehension. Examples include:
- Lines 36–37: awkward phrasing
- Line 41: “MSCs” is plural, but followed by singular verbs
- Lines 50–53: unclear sentence structure
- Line 207–208: rephrase for clarity
- Line 226–227: possible redundancy
- Line 226–235: clarify the values in parentheses – what does the “±” represent?
- Ensure consistent terminology throughout the manuscript. For instance:
- Use only "LAA" after first mention; replace "AA" elsewhere.
- “MSC” vs “MSCs”: maintain correct singular/plural usage.
- Abbreviations such as ROS should only be defined once.
- The mechanistic link between LAA stimulation and anti-aging effects is not fully explained—consider including a schematic or graphical summary.
- Introduction
- Line 63: "AA" should be consistently replaced with “LAA” after abbreviation is introduced.
- Materials and Methods
- Line 112: When ISCT is first mentioned, please explain the acronym and provide the full name.
- Line 160: Ensure abbreviations are defined once and consistently used throughout (e.g., use only LAA).
- LAA concentrations used in the experiments are not clearly justified. A brief rationale based on prior studies would strengthen the scientific basis.
- Figures and Legends
- Figures 1 and 2 are not referenced in the text. Please ensure all figures are cited where appropriate.
- Figure legends (e.g., for Figures 4 and 7) require expansion. Include more explanation to aid reader understanding.
- Clarify statistical indicators in figure legends. For instance, what do asterisks (*) in Figure 7 represent? Which p-values are significant?
- Results Section
- Clarify line 207–208: the sentence is grammatically unclear.
- Lines 226–235: clarify the use of mean ± SD and ensure it is consistent with statistical reporting standards.
- Discussion
- Maintain consistent terminology. Avoid repeating definitions (e.g., "MSC", "ROS").
- Line 299: please cite the full reference (e.g., “a study by Segeritz et al.”).
Must be improved
Reviewer 4 Report
Comments and Suggestions for Authors
In the manuscript (ID: cimb-3636173), the authors researched the correlation of L-ascorbic acid supplementation on adipose-tissue mesenchymal stem cells (AT-MSCs) secretion, and consider this as a potential method for treating skin aging. In general, the research meets the requirements of Current Issues in Molecular Biology (CIMB). Therefore, I think this manuscript can be published in Current Issues in Molecular Biology (CIMB) after a major revision.
(1) Title: The meaning expressed in the title is not very reasonable. It is suggested that the author revise it. Change “The Correlation of L-Ascorbic Acid Supplementation on Adipose-Tissue Mesenchymal Stem Cells (AT-MSCs) Secretion as a Potential Treatment for Skin Aging” to “L-Ascorbic Acid Supplementation can Serve as a Potential Treatment for Skin Aging Because It Can Regulate the Secretion of Adipose-Tissue Mesenchymal Stem Cells (AT-MSCs)”.
(2) Line 19 and 23: " in vitro” should be italic. In addition, there are similar errors in other parts of the manuscript, and authors are advised to check the whole manuscript carefully and correct these minor errors.
(3) Line 24: Please provide the full name of IGF-1, TGF-β1, and IL-6. When an abbreviation appears in the manuscript, write its full name first, and the abbreviation is written after the full name in parentheses. Subsequently, use the abbreviation consistently and do not write out the full term again.
(4) Keywords: Change “Adipose Tissue Mesenchymal Stem Cells (AT-MSCs)” to “Adipose tissue mesenchymal stem cells (AT-MSCs)”.
(5) Keywords: Change “L-Ascorbic Acid (LAA)” to “L-ascorbic acid (LAA)”.
(6) In the introduction, the authors lack a deep review of the research on the treatment of skin aging with natural active substances. At present, there are some studies on treatment of skin aging with polyphenols, collagen, and bioactive peptides, such as protective effects on H2O2-induced human skin fibroblasts of antioxidant peptides from Volvariella volvacea, chemical characterization of honeysuckle polyphenols and their alleviating function on ultraviolet B-damaged HaCaT cells by modulating the Nrf2/NF-κB signaling pathways, effects of Collagen supplements on skin aging, cytoprotection on ultraviolet-A injured human skin fibroblasts of antioxidant peptides from Skipjack tuna (Katsuwonus pelamis) skins, protective function on UVB-irradiated HaCaT cells through antioxidant and anti-apoptotic mechanisms of Bioactive Peptides from Skipjack tuna cardiac arterial bulbs, etc. It is suggested that the authors systematically review of these polyphenols, collagen, and bioactive peptides to further explain the importance and significance of this research.
(7) 2. Materials and Methods: “Materials and reagents” are omitted in this part. Suggest authors to add the key reagents. In addition, please check carefully and add information on relevant Materials and reagents. It is of utmost importance this is clarified and more detailed to allow replication.
(8) 2. Materials and Methods: Each method written in the manuscript should include the reference used, which allows the reader to access the original experimental methods.
(9) Line 160-161: Change “L-Ascorbic Acid (LAA)” to “LAA”.
(10) Line 173 and 203: “2.5 Statistical Analysis” and “3.2 LAA treatment in AT-MSCs improved cell morphology and enhanced the number of live cells without affecting cell viability”. It is recommended that authors carefully read the submission requirements of Current Issues in Molecular Biology (CIMB), confirm whether the first letter of each word in the title needs to be capitalized, and unify the writing of the words in the title.
(11) Line 186-187: …which are 186 99.8% of CD73, 95% of CD90, 62.7% of CD105… It is suggested to use the conventional format "mean ± SD%" for data presentation. Review the entire text and make corrections accordingly.
(12) Line 209: p < 0.05. "p” should be italic. In addition, there are similar errors in other parts of the manuscript, and authors are advised to check the whole manuscript carefully and correct these minor errors.
(13) Line 206-235: The author should analyze the obtained data instead of simply listing them here.
Comments on the Quality of English LanguageThe English could be improved to more clearly express the research.
Round 2
Reviewer 4 Report
Comments and Suggestions for Authors
The authors have carefully revised the manuscript (ID: cimb-3636173) and the quality of the manuscript has been improved accordingly. Therefore, I think that the manuscript can be accepted for publication in Current Issues in Molecular Biology (CIMB).